# Distinct Effects of Wheat and Black Bean Consumption on Postprandial Vascular Responses in People with Arterial Stiffness: A Pilot Randomized Cross-Over Study

**DOI:** 10.3390/nu17071159

**Published:** 2025-03-27

**Authors:** Peter Zahradka, Danielle Perera, Jordan Charney, Carla G. Taylor

**Affiliations:** 1Canadian Centre for Agri-Food Research in Health and Medicine, St. Boniface Albrechtsen Research Centre, Winnipeg, MB R2H 2A6, Canada; pzahradka@sbrc.ca (P.Z.); dperera@sbrc.ca (D.P.); jcharney@sbrc.ca (J.C.); 2Department of Physiology and Pathophysiology, University of Manitoba, Winnipeg, MB R3E 0J9, Canada; 3Department of Foods and Human Nutritional Sciences, University of Manitoba, Winnipeg, MB R3T 2N2, Canada

**Keywords:** black beans, wheat, rice, pulse wave velocity, augmentation index, blood pressure, postprandial response

## Abstract

**Background/Objective:** Postprandial vascular responses impact vascular health. This study investigated whether eating pulses or whole grains can acutely relax blood vessels in people with arterial stiffness. **Methods:** A single-blinded, controlled randomized cross-over clinical trial was conducted to compare the effects of pulses (¾ cup black beans) versus whole grains (¾ cup whole wheat kernels, also known as wheat berries) versus white rice (¾ cup) on postprandial vascular responses in males and females with established arterial stiffness (n = 9, 3M/6F, 50–64 years old). Peripheral and central hemodynamic measurements were obtained non-invasively prior to and 2 h after food consumption and were compared by t-test within a food type. **Results:** Peripheral and central systolic blood pressure was increased (4%) after eating white rice but not after the consumption of wheat or beans. A marked decline in augmentation index at 75 bpm (arterial stiffness) from 26.1 ± 3.6% to 16.2 ± 2.0% was observed 2 h after eating whole wheat but not beans or white rice. All foods slightly decreased heart rate at 2 h but had limited effects on other parameters of circulatory or heart health. **Conclusions:** Eating whole wheat or beans acutely improved overall vascular and heart health when compared to white rice. The effects of wheat and beans were distinct, with whole wheat having a major positive effect on blood vessel stiffness. The findings suggest that regular inclusion of both whole wheat and beans in the diet would be beneficial for improving cardiovascular health in persons exhibiting signs of arterial dysfunction, thus providing a potential therapeutic benefit for individuals who are at risk of heart attack and stroke. The study was registered (NCT05818358) on ClinicalTrials.gov.

## 1. Introduction

Cardiovascular disease (CVD) is the leading cause of morbidity and mortality worldwide, accounting for approximately one third of all deaths annually [1,2]. A key precursor to CVD is age-related arterial dysfunction, characterized by increased stiffness of the arteries and impaired function of the arterial wall (endothelial function) [3]. Thus, arterial stiffness is an important marker of vascular damage and a strong predictor for assessing the risk of cardiovascular events such as heart attacks and strokes [1,4].

The concept of arterial stiffening refers to a change in the structural and functional properties of the arterial wall, primarily the loss of elasticity [5]. The generally accepted measurement of arterial stiffness is pulse wave velocity (PWV), which non-invasively records the time it takes for a pulse pressure wave, generated by the ejection of blood from the heart, to travel a specific distance [6,7]. Fundamentally, a pulse wave travels faster in arteries that are stiff compared to those with greater elasticity. Consequently, as large arteries stiffen due to age or disease, blood pressure and flow increase, which affects the function of smaller arteries such as those within target organs (e.g., brain, kidney, heart, eyes) [8]. Therefore, intervention to lessen arterial stiffness is important for preventing further progression of CVD and additional damage to other target organs.

Current preventative or management strategies for arterial stiffness include pharmacological therapies and lifestyle modifications, of which dietary intervention remains a cornerstone due to its safety and cost-effectiveness [9]. Various studies have shown evidence of dietary patterns affecting arterial stiffness, with healthier dietary patterns (e.g., vegetable-rich, containing pulses or whole grains) associated with lower levels of arterial stiffness and/or improved vascular health [10,11,12,13,14]. While long-term dietary patterns are an established factor for CVD risk, there is also emerging evidence of postprandial vascular responses playing a pivotal role in the direction of our vascular health. Since most time during the day and evening is spent in the postprandial (i.e., fed) state, how blood vessels immediately respond to our food choices can be highly impactful over time. It is suggested that regularly consuming foods that induce post-meal vasoconstriction (e.g., high-fat fast foods) [15,16] may lead to arterial stiffness through impairment of endothelium-dependent vasodilation [17], a process that can contribute to the development of CVD over the lifespan by promoting arterial remodeling [18]. Indeed, changes in arterial stiffness based on PWV measurements, the most direct indication of arterial stiffness, have been detected within two hours after ingestion of glucose [19]. Conversely, consuming foods that promote the relaxation of blood vessels may be beneficial for long-term vascular health [20,21], and their identification is being actively investigated.

We have previously reported that consuming red kidney beans or black beans relaxes blood vessels in healthy individuals shortly after the beans are consumed, representing a positive effect of darker-colored beans on the postprandial vascular response [20]. From this, it is speculated that certain bioactive compounds (phytochemicals) contained in the seeds of these beans are likely responsible. In addition to pulses, there is published evidence indicating that whole grains may have similar relaxation effects on blood vessels due to their phytochemical constituents [22].

To date, no studies have compared the post-meal vascular response of beans to another phytochemical-rich food product, nor has any study examined post-meal vascular responses in people with established disease. As our previous postprandial vascular response study was in healthy adults, our goal was to apply our findings to individuals with arterial stiffness. Thus, this is the first study to examine the postprandial vascular effects of beans and whole grains in a population with asymptomatic CVD. Furthermore, this study employed the intact kernel as various processing techniques (e.g., milling) remove valuable phenolic compounds and other constituents present in the bran layer (outer husk).

The purpose of this research was to examine whether eating beans or whole grains can acutely relax blood vessels in people with arterial stiffness. Both food products (beans and wheat) were compared to white rice, which provides an equivalent number of calories but a lesser amount and types of phytochemicals, micronutrients, and fiber. The food products were provided to the participants as the first meal of the day (breakfast), and vascular responses were measured 2 h following the first bite.

## 2. Methods

### 2.1. Study Design

This was a single-site, single-blinded, randomized, controlled cross-over study conducted at the Asper Clinical Research Institute, St. Boniface Hospital, Winnipeg, Canada. The study was designed to compare the effects of black beans or whole grain (whole wheat as intact kernels, also known as wheat berries) or white rice (isocaloric comparator) at breakfast on postprandial (2 h) vascular responses in males and females with arterial stiffness. Recruitment consisted of a total of 10 participants with arterial stiffness as defined by a brachial-ankle pulse wave velocity (baPWV) of greater than 1400 cm/s [23]. The sample size for this pilot study was based on our previous study in healthy individuals, where n = 10 was sufficient to reveal differences in postprandial vascular responses after consumption of different bean varieties [20]. Participants were recruited from the local community using a variety of advertising approaches. Participants provided informed consent before any study procedures were conducted. They attended an in-person screening visit to provide a fasting blood sample and to assess the presence of arterial stiffness (by baPWV) to determine eligibility. Eligible participants were scheduled for three study visits to assess postprandial blood vessel function. The aortic augmentation index normalized to 75 heartbeats per minute (AIx75) was the primary endpoint. Other peripheral and central hemodynamic parameters were secondary endpoints. The study visits were conducted between July 2023 and March 2024. The study protocol was approved by the University of Manitoba Research Biomedical Ethics Board (HS25221) and the St. Boniface Hospital Research Review Committee (RRC/2021/2043). The study was registered (NCT05818358) on ClinicalTrials.gov.

### 2.2. Study Population

Participants met the following criteria to be eligible for participation in the study: male, or non-pregnant, non-lactating female, 45 to 65 years of age; baPWV > 1400 cm/s on at least one side (indicative of clinically significant arterial stiffness [23]); plasma creatinine ≤ 265 µmol/L; aspartate aminotransferase (AST) < 160 U/L, and alanine aminotransferase (ALT) < 150 U/L; glycated hemoglobin ≤ 6.5%; LDL-cholesterol < 5 mmol/L; stable regimen if taking vitamin and mineral/dietary/herbal supplements for 1 month prior to starting the study and while participating in the study; had not donated blood or blood products (e.g., platelets) during the 2 months prior to starting the study or while participating in the study; not participating in another dietary intervention trial for the past month and not starting another dietary intervention trial for the duration of the study; willing to comply with the protocol requirements and procedures; willing to provide informed consent.

Participants were excluded for the following: had experienced a cardiovascular event (e.g., heart attack, stroke) or had a surgical procedure for cardiovascular disease (e.g., bypass, stent), presence of clinically diagnosed cardiac arrhythmia or valve stenosis or peripheral arterial disease (PAD), chronic renal disease, liver disease (with exception of fatty liver), lung disease, rheumatoid arthritis, immune disorders or diseases (e.g., multiple sclerosis, leukemia), cancer in the previous 5 years, neurological disorders, or gastrointestinal disorders; taking vasoactive medications (e.g., angiotensin converting enzyme inhibitors, α-receptor blockers, advanced glycation end-product breakers, thiazolidinediones, beta-blockers, statins, insulin, etc.); blood pressure > 160 mmHg systolic and/or >100 mmHg diastolic; history of gastrointestinal reactions or allergies to beans, bean flour, wheat, wheat flour, gluten, or rice; current (within the past 30 days) bacterial, viral or fungal infection; bleeding disorders; amputations of upper or lower extremities on both sides; any acute medical condition or surgical intervention within the past 3 months; drug and/or alcohol abuse; psychological disorder (s); inability to fast overnight; unable to take prescribed medication without food; and unable to obtain vascular function measurements and/or blood sample at the screening or first study visits.

### 2.3. Food Preparation

The food items for this study were prepared in the Clinical Food Preparation Kitchen at the Asper Clinical Research Institute, following standard food handling procedures. All 3 items were prepared in a commercially available vegetable broth for better palatability. Dry black beans (sourced by Pulse Canada, Winnipeg, MB, Canada) were rinsed and then cooked in Campbell’s Ready To Use Vegetable Broth (1:4.5 *wt*/*wt*; Mississauga, ON, Canada) for 75 min. The cooked beans were then rinsed. The ¾ cup volume was weighed 5 times and averaged 128.5 g. Dry wheat berries (Bulk Barn, Winnipeg, MB, Canada) were rinsed and then cooked in Campbell’s Ready To Use Vegetable Broth (1:3 *wt*/*wt*) for 60 min. The cooked wheat berries were then rinsed. The ¾ cup volume was weighed 5 times and averaged 120.7 g. White basmati rice (Uncle Ben’s, Houston, TX, USA) was rinsed and then cooked in Campbell’s Ready To Use Vegetable Broth (1:1.75 *wt*/*wt*) in a rice cooker. The ¾ cup volume was weighed 5 times and averaged 95.0 g. All samples were stored at −20 °C until used. The day/evening before a participant’s visit, the food item was thawed in the refrigerator. Immediately before serving, the food item was heated for 45 s in a microwave.

### 2.4. Randomization Procedures and Blinding

A list of unique product code numbers (randomization codes) was prepared by a statistician to ensure participants received each of the test products (beans, whole wheat, or white rice) in a random order. Participants were enrolled sequentially in the study. After randomization, the study coordinator was responsible for ensuring that the assigned randomization codes were applied to all study documentation for that particular participant and for dispensing the test products that matched the randomization code.

The participants were not blinded to the order and the type of food product they received at a study visit because they could identify the food products by taste. However, all food samples were assigned a randomization code to ensure the study team and laboratory personnel were blinded to the identity of the test products until the end of the study to reduce potential bias during data collection and analysis.

### 2.5. Study Visits and Assessments

#### 2.5.1. Screening Visit

After participants had reviewed and signed the informed consent form, demographic information, a complete medical history, including clinically diagnosed diseases, and a physical assessment [blood pressure, height, weight, calculated body mass index (BMI), waist circumference, and body composition by bioelectrical impedance (InBody 570, InBody Canada, Ottawa, ON, Canada)] were obtained. The presence of arterial stiffness via baPWV and peripheral arterial disease (PAD) via ankle-brachial index (ABI) was determined with the VP-1000 (OMRON Healthcare, Hoffman Estates, IL, USA) [5]. Blood pressure was measured with the automated BPTru oscillometric blood pressure monitor (VSM MedTech, Coquitlam, BC, Canada), as previously described [5]. A fasting blood sample was obtained for serum biochemistry (total cholesterol, LDL-cholesterol, HDL-cholesterol, triglycerides, creatinine, AST, ALT, and glycated hemoglobin) and analyzed by the Biochemistry Lab at St. Boniface Hospital (Shared Health, Winnipeg, MB, Canada). The information obtained at the screening visit was used to determine eligibility for the study based on the inclusion and exclusion criteria.

#### 2.5.2. Study Visits

Eligible participants attended 3 study visits, a minimum of 6 days apart, that were scheduled for the morning. Participants arrived fasted (no food or beverage consumption except water 12 h before the visit). At each visit, one of the following products was consumed: (i) ¾ cup of cooked black beans; (ii) ¾ cup of cooked whole wheat (wheat berries); or (iii) ¾ cup of cooked white rice. Participants were provided with 100 mL of water with the food item and were allowed to drink an additional 150 mL of water during the 2 h test period. The participants were required to consume the study product within 10 min. The study coordinator observed study food consumption and recorded the time. AIx75 (primary endpoint) and other hemodynamic and pulse wave parameters were measured with a SphygmoCor XCEL device (AtCor Medical Pty, Sydney, Australia) [24] at time point 0 (prior to consumption, T0) and 2 h (T120) after the first bite of the food product.

### 2.6. Statistical Methods

Outliers were defined as data points greater than 2.5× the standard deviation from the mean, resulting in the exclusion of data from one participant in the statistical analyses. Data were analyzed by *t*-test (using Microsoft Excel 2019, version 1808, Microsoft Office, Redmond, WA, USA) to compare T0 and T120 values within a food type (i.e., wheat, bean, white rice). The time to consume the 3 food types was analyzed by ANOVA, followed by Duncan’s multiple range test, with Statistical Analysis Software (SAS version 9.4, SAS Institute Inc., Cary, NC, USA). Statistical significance was set at *p* ≤ 0.05. Principal component analysis (PCA) was performed with XLSTAT (version 2023.2.0, Addinsoft, New York, NY, USA).

## 3. Results

### 3.1. Participant Characteristics

Eleven participants completed all three study visits (Figure 1). However, the pulse wave data obtained from one participant were of insufficient quality to provide results, and another participant was removed as many of their values for the different parameters were outliers. Thus, the data from nine participants are included in the analysis of this pilot study.

The characteristics of the nine participants (three males, six females) are summarized in Table 1. The average age was 57 ± 6 years (mean ± SD) and ranged from 50 to 64 years of age. All participants had an elevated risk of cardiovascular disease based on their body mass index (all above 25 kg/m^2^); however, their serum biochemistry values for LDL-cholesterol (normal < 4 mmol/L) and triglycerides (normal < 3 mmol/L) ranged from normal to high, as did their blood pressure (normal < 120 mmHg systolic, <80 mmHg diastolic). ABI was normal (>0.9), indicating an absence of PAD. The participants did not have renal or liver disease or diabetes based on values for creatinine, ALT, AST, and hemoglobin A1c. For the purpose of this study, the main selection criterion was an elevated baPWV, a measure of arterial stiffness. Specifically, participants were eligible if they had a baPWV of ≥1400 cm/s on at least one side, either left or right, as determined with the VP-1000 instrument. This requirement was achieved, as the minimum left-side baPWV was 1445 cm/s. Eligible participants proceeded to the study visits where they consumed the three food articles on different days, and the data obtained were analyzed for changes in endpoints associated with vascular function. It took longer for the participants to consume the wheat (8.0 ± 0.6 min; mean ± SEM) and beans (7.3 ± 0.5 min) compared to rice (4.9 ± 0.5 min; *p* < 0.05).

### 3.2. Peripheral Hemodynamic Parameters

Blood pressure was measured because it can respond to the type of food that is consumed, resulting in a transient change that may last a few hours after eating [20]. In this study, eating white rice led to a significant elevation of 6.2 mmHg in peripheral systolic pressure, but there was no significant change when either wheat or beans were eaten (Table 2). Peripheral diastolic pressure was increased by 2.9 mmHg by black beans. Peripheral pulse pressure, the difference between the systolic and diastolic pressures, was increased by 6.7 mmHg by wheat and 5.4 mmHg with white rice. No significant changes were noted for the mean arterial pressure (⅓ SBP + ⅔ DBP).

### 3.3. Central Hemodynamic Parameters

Since the aorta has the highest pressures along the circulatory tree, it is typically more sensitive to higher pressures and begins to deteriorate much sooner than the peripheral blood vessels. For this reason, determining the central hemodynamic parameters offers greater information about cardiovascular disease state than the blood pressure at the brachial artery [25]. The pressure on the aorta can be determined from the shape of the pulse wave as it traverses the brachial artery. Specifically, algorithms use the height of the pulse wave to estimate the aortic (central) pressure. Analysis of the pulse wave also provides other parameters related to the health of the circulatory system and the heart [26,27]. As was seen with the peripheral parameters, white rice consumption caused a significant increase of 5.7 mmHg in aortic systolic pressure while neither wheat nor beans had an effect (Table 3). The pulse pressure was also higher after consumption of white rice (4.5 mmHg) but not with wheat or beans.

A key feature of pulse wave analysis is the ability to determine the augmentation index (AIx), which refers to the force that the heart must pump against. AIx represents the increase in blood pressure that occurs when the pulse wave initiated by one contraction overlaps with the reflection of the pulse wave from the previous contraction [23,27]. The reflected wave, which occurs as a result of vessel narrowing and branching, augments the pressure above that of the systolic pressure, and the magnitude of the augmented wave is related directly to the stiffness of the vessel wall. In this study, AIx was unchanged by consumption of the foods, but when the values were normalized to a constant heart rate of 75 beats per minute, it revealed that wheat lowered AIx75 from 26.1% to 16.2% (Figure 2). This information indicates arterial stiffness was reduced considerably in the participants after consuming the wheat, but not by beans or white rice.

Additional hemodynamic parameters that were significantly changed by the foods include the pressure-time index (PTI) in diastole, end-systolic pressure, and mean arterial pressure in systole (MAPsys) (Table 4). Each of these increased in response to white rice, but not wheat or beans. The end systolic pressure is an indication of the force generated by the heart at the end point of contraction (systole) [28], whereas MAPsys defines the mean arterial pressure during contraction. In contrast, PTI is an indication of how much energy is being supplied to the heart. Additionally, the P1 height was increased with both wheat and white rice indicating that both foods caused an elevation in the pulse wave pressure. Other hemodynamic parameters, namely aortic augmentation pressure, Buckberg SEVR (subendocardial viability ratio), PTI systole, and MAP diastole did not change. Likewise, wave reflection parameters (forward pulse height, reflected pulse height, reflection magnitude), which are indicators of the dynamics of the forward pulse wave and the reflected pulse wave when considered separately, also remained unchanged (Table 5).

Several parameters associated with heart function were affected by the foods [24]. Specifically, all foods significantly decreased heart rate, while the length of the average pulse and the ejection duration were increased by both wheat and beans (although when expressed as a percentage, only beans affected the duration), and beans increased the duration of the pulse from the start to the peak at systole (Aortic T2). In all cases, the changes are indicative of better heart function since the longer ejection time typically means more blood is being expelled from the heart with each beat.

### 3.4. Principal Component Analysis

Principal component analysis was performed by inputting the AIx75 delta (T120—T0) for each food and the anthropometric and serum biochemistry parameters of the participants (Figure 3). The principal component analysis differentiated the three food items based on the AIx75 delta response. The F1 component explained 97.85% of the variance, with waist circumference, percent body fat, and AIx75 delta for wheat having the highest factor scores for F1 (9.18, 2.42, and −2.54, respectively). The clustering of serum cholesterol, triglycerides, hemoglobin A1c, and the skeletal muscle index with the delta AIx75 for white rice near the origin indicated these parameters had a minimal effect on the response to white rice.

## 4. Discussion

The relationship between food and disease is well recognized when considering the effects of excessive food consumption [29]. This is especially true when the amount of fat exceeds the storage capacity of the adipose depots, resulting in the deposition of fat in non-adipose tissues, such as the liver, pancreas, muscle, and heart [30]. When this occurs, a variety of metabolic disorders can emerge, such as fatty liver disease and diabetes, which in turn serve as the foundation for the development of cardiovascular disease [30,31]. Interestingly, it has also been shown that the regular consumption of foods that negatively impact the blood vessels in the postprandial period can promote cardiovascular disease even in the absence of a metabolic imbalance. These negative effects of eating are usually related to the amount and type of fat or carbohydrate in the food [15,32,33,34,35]. The effects on blood vessels can be profound, affecting a number of functions, including arterial stiffness [16]. At the same time, it has been reported that certain foods or food ingredients can block the negative actions of fat and carbohydrates on blood vessels or have beneficial effects on the blood vessels directly [20,36,37]. Indeed, clinical trials exploring the effects of pulses and protein sources on postprandial arterial stiffness have been completed or are in the planning stages [20,38]. The novel contribution of the present study is the distinct effects of wheat and beans on postprandial vascular responses, with both affecting blood pressure, while whole wheat had a positive effect on reducing blood vessel stiffness 2 h post-consumption, as indicated by the significant decline in augmentation index (AIx75).

A major outcome of this study was the significant decline in augmentation index (AIx75) obtained with wheat consumption, but not with beans or white rice (Figure 2). This finding agrees with earlier studies that reported a negative correlation between whole grain consumption and arterial stiffness [10,13]; however, neither of these publications examined arterial stiffness in the postprandial state. The observation that eating beans did not lead to a decrease in arterial stiffness also agrees with a prior publication, and in this case, it did look at the effects postprandially [20]. AIx is a measure of arterial stiffness [23,39], and the reduction of AIx75 from 26.1% to 16.2% obtained with wheat represents a very significant change. Interestingly, the physiological parameters that influence AIx were not obviously affected by eating wheat. Although augmentation pressure (AP) was slightly decreased after eating wheat, and this value was slightly higher when beans and white rice were consumed, they were not statistically significant. The change in AP may be related to the higher P1 height, as this value is employed in calculating AIx [23,24,27]. Since the product eaten by the participants was intact whole wheat kernels, it will be of interest to determine whether processing would lead to a loss in the ability to affect AIx. This possibility is raised because in a previous publication, we reported that a fine-milled bean flour (versus coarse-milled bean flour) was less effective in lowering LDL-cholesterol [40].

It is noteworthy that whole wheat consumption reduced AIx75 from 26.1% to 16.2% in middle-aged individuals with pre-existing asymptomatic CVD (i.e., presence of arterial stiffness but no previous cardiovascular event, no cardiovascular surgery, and not taking statins, antihypertensive or other vasoactive medications) who were overweight or obese based on body mass index. Principal component analysis revealed that waist circumference, an indicator of visceral fat deposition, was a major factor in the postprandial AIx75 response. The serum metabolic factors related to cholesterol, triglycerides, glycemic control, and skeletal muscle index were clustered with the delta AIx75 for white rice, but their location near the origin of the plot suggests these metabolic factors had a minor effect on the postprandial response and that they were not impacting the postprandial response to beans and whole wheat. Future studies will need to investigate the relationships among the food constituents (including phytochemicals), the plasma metabolomic profile pre- and post-consumption, and the postprandial vascular responses to identify the molecules/metabolites from the foods consumed and the vascular response that can elicit the positive changes in AIx75. This also needs to be investigated in the context of how an individual’s visceral fat deposition impacts their postprandial vascular response to specific foods.

Another main outcome of this study is that peripheral (brachial) systolic blood pressure (Table 2) and central (aortic) systolic pressure (Table 3) were increased by white rice but not by wheat or beans. A higher peripheral diastolic pressure was observed with beans, but this did not translate into an increase centrally. Repeated increases in blood pressure, even if they are transient, can lead to arterial remodeling [41]. A change in the structure of a vessel to resist pressure is the first step in making the vessel stiff, and this represents the first stage in the development of atherosclerosis [42,43]. Not all foods can cause an increase in blood pressure. As seen in this study, wheat and pulses do not. This finding suggests the effects of wheat and beans may be neutral and unlikely to promote cardiovascular disease. Previous studies have shown that certain food components, such as simple sugars and saturated fat, can elevate blood pressure in the postprandial period [44]. Therefore, the consumption of processed foods (e.g., the addition of sugar and/or saturated fat, and the removal of the outer husk containing phytochemicals and fiber) versus whole foods may be a determining factor in whether a food is capable of increasing blood pressure or not.

Interestingly, the inability to decrease blood pressure may provide a secondary benefit. It has been reported that a variety of foods can cause a reduction in blood pressure, and that this action is enhanced by advanced age. This postprandial hypotension, which is closely related to orthostatic hypotension, is a major cause of morbidity and mortality in the elderly and predisposes individuals to cardiovascular disease [45,46,47]. For these reasons, there is considerable interest in identifying pharmaceuticals that can prevent it [48]. Since it appears that most macronutrients (carbohydrates, fats, proteins) can induce postprandial hypotension [49], identifying foods that can prevent this condition may provide a novel route for therapeutic treatment [50]. It is probably noteworthy that the average age of the participants in this study was 57 years, and this age is likely low relative to those most susceptible to postprandial hypotension.

In addition to blood pressure, pulse pressure (both peripheral and central, Table 2 and Table 3) was also elevated in response to white rice. Peripheral pulse pressure was also elevated non-significantly after consuming wheat, but not in response to beans; this higher peripheral pulse pressure, however, did not affect central pulse pressure for wheat. In contrast, mean arterial pressure was unaffected by any of the treatments. While pulse pressure and mean arterial pressure (MAP) are not usually considered when evaluating hemodynamic parameters, they both have physiological relevance [51,52], with peripheral pulse pressure representing the force generated when the heart contracts while MAP is a measure of the resistance to blood flow in the circulatory system. A pulse pressure higher than 50 mmHg or lower than 30 mmHg is associated with an increased risk of CVD [51]. Our study participants exhibited pulse pressure in the normal range. However, a transient increase in peripheral pulse pressure could lead to arterial remodeling, as described above for systolic pressure. While MAP was unaffected by the foods, a high MAP (>100 mmHg) indicates that blood flow is being impeded, while a low MAP (<60 mmHg) suggests that blood flow is insufficient to reach all the vital organs [52]. In this study, the average MAP was above the threshold of 100 mmHg, suggesting their circulatory system is resisting blood flow. The high MAP is likely due to all of the participants having arterial stiffness, based on the high pulse wave velocity that was the primary inclusion criterion. Arterial stiffness indicates narrowing and hardening of blood vessels due to plaque deposition [23,51], and thus impeded blood flow would be expected.

All of the food items increased heart rate (Table 3 and Table 4). This appears to be a common effect associated with eating [53]. Both wheat and beans also increased ejection duration (in ms) and period, which are related to greater efficiency of heart pumping [54], as is ejection duration (in %), which was only altered by beans. The pressure-time index, end systolic pressure, and MAP (systolic) were only elevated by white rice. The pressure-time index is related to the work the heart is performing while beating [24], which may explain why the end pressure at systole is higher. Interestingly, no wave reflection parameters were affected by the foods (Table 5). The pulse wave is reflected at sites where the arterial tree branches and narrows. When the reflected wave overlaps with a subsequent pulse wave, the pressure is augmented, and this is used to calculate AIx [23,24,27]. However, the speed of the reflected wave is an independent index of arterial stiffness, as greater stiffness results in a quicker return [51]. Given the magnitude of change in AIx with wheat, the lack of effect on wave reflection is surprising. However, this difference may provide some insight into how wheat decreases arterial stiffness once we obtain a better understanding of how quickly such parameters might change in response to treatment.

Postprandial hyperglycemia (“glucose spikes”) is recognized as an independent risk factor for CVD in individuals with and without cardiometabolic disease or diabetes [55,56,57]. Elevating blood glucose to 10 mmol/L for 3 h via an euglycemic hyperglycemic clamp impairs endothelial dysfunction assessed by flow-mediated dilation (FMD) [58]. However, it is recognized that the glycemic index of foods and the dietary glycemic load have an impact on CVD risk beyond that of simple sugars [59]. Thus, it is important to consider the possibility that differences in eating speed, the type of food, and their influence on postprandial blood glucose concentrations could be contributing factors to the postprandial responses in the present study. However, the differences in eating speed between rice and wheat (3.1 min) and rice and beans (2.4 min) are unlikely to explain the effects of rice on blood pressure parameters at 2 h post-consumption. There was no statistical difference in the eating time between wheat (8.0 ± 0.6 min) and beans (7.3 ± 0.5 min), yet wheat had a significant effect on lowering AIx75, an indicator of arterial stiffness (Figure 2). Although we did not monitor blood glucose, there were no differences in blood glucose concentrations between beans and rice at 1 h or 2 h post-consumption in our previous study comparing postprandial vascular responses of different bean types and rice in healthy individuals [20]. Wheat and beans have a low glycemic index (<50), and the comparator food, rice, has a lower glycemic index than a glucose solution [59], thus large differences in blood glucose would not be expected. It is noteworthy that most studies demonstrating negative effects of post-challenge hyperglycemia on FMD have been conducted with a glucose solution (i.e., oral glucose tolerance testing) [57], whereas the present study investigated the effects of whole foods. The glycemic properties of foods are influenced by a number of factors [59], and a recent study with defined meal compositions has shown that the combination of low glycemic index and high cereal fiber increases postprandial FMD [60]. In the present study, whole wheat, which has a low glycemic index and high cereal fiber, reduced postprandial arterial stiffness. Participant characteristics are another factor to consider when interpreting postprandial responses. Participants in the present study had metabolic risk factors (Table 1), but none were taking glucose- or lipid-lowering medications. It is possible that clinical characteristics such as basal metabolic rate and muscle mass could be impacting insulin and glucose responses, and these factors need to be addressed in future studies. In addition to an oral glucose challenge, high-fat meals have negative effects on postprandial endothelial dysfunction [17], however, it is important to keep in mind that endothelial dysfunction is distinct from PWV and does not equate to arterial stiffness [5]. Rather, chronically repeated postprandial endothelial dysfunction over months and years results in remodeling of the vasculature, which culminates in arterial stiffness [18]. The present study demonstrates that the consumption of wheat reduces arterial stiffness postprandially, and further studies are needed to evaluate the long-term effects on arterial remodeling.

A strength of this study was the investigation of the acute effects of bean and whole grain consumption on blood vessel health in middle-aged individuals with established CVD, specifically those at higher risk of heart attack and stroke, versus studies that focus on healthy young individuals. An objective criterion, baPWV > 1400 cm/s [23], was employed to define participants with arterial stiffness and who had established but asymptomatic CVD. This pilot study demonstrated the application of a non-invasive approach and equipment for assessing and differentiating the postprandial vascular responses elicited by different foods. This study provides data for calculating sample sizes in future studies. The results support the positive acute postprandial effects of whole wheat and beans consumed as the sole food item; however, further studies are needed to evaluate their effects within mixed meals, whether they can mitigate the effects of food ingredients or processing approaches with deleterious effects, and whether habitual consumption can maintain or improve blood vessel health in individuals with established CVD. A limitation of the current study is the lack of data for blood glucose and other metabolic parameters at baseline and during the postprandial phase, and these assessments are recommended for future studies.

## 5. Conclusions

Eating whole wheat or beans acutely improved overall vascular and heart health when compared to white rice. The effects of wheat and beans were distinct, with both affecting blood pressure while whole wheat had a major positive effect, reducing postprandial AIx75, an indicator of arterial stiffness. The results support the concept that habitual inclusion of beans and whole wheat in the diet may mitigate the deleterious effects of arterial stiffness, thus further reducing the risk for CVD. This investigation also provides the foundation for future long-term studies identifying the dose and duration of bean (and other pulses) and/or whole grain (wheat and other grains) consumption for improving the structure and function of blood vessels in individuals with preexisting disease, as well as to determine if their inclusion in the diet can mitigate the actions of foods known to negatively affect cardiovascular health.

## Figures and Tables

**Figure 1 nutrients-17-01159-f001:**
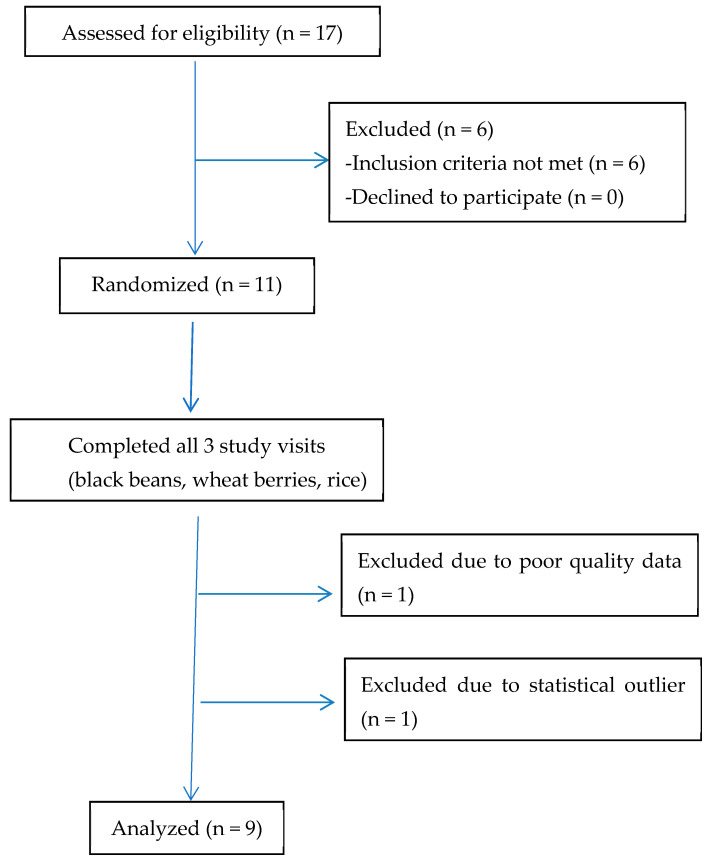
CONSORT diagram.

**Figure 2 nutrients-17-01159-f002:**
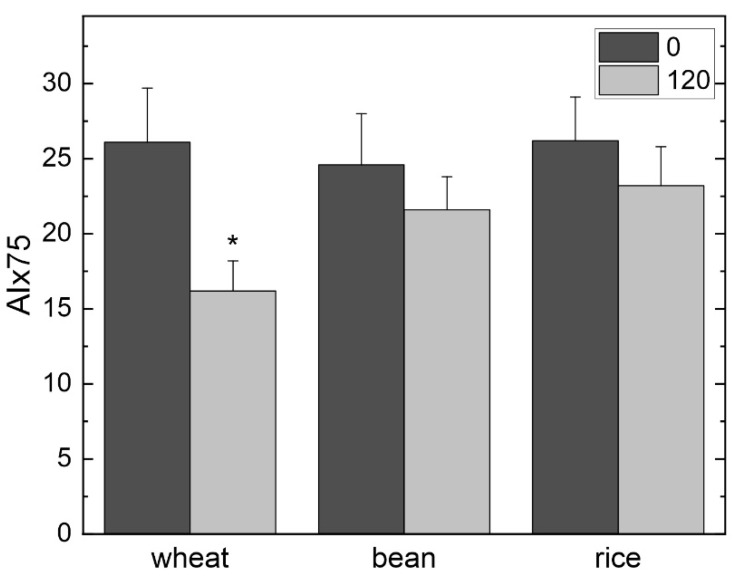
Augmentation index normalized to a heart rate of 75 beats/minute (AIx75) before (0 min) and after (120 min) consumption of wheat, beans, or rice. The asterisk (*) indicates a statistically significant (*p* < 0.05) change between 0 and 120 min within a food.

**Figure 3 nutrients-17-01159-f003:**
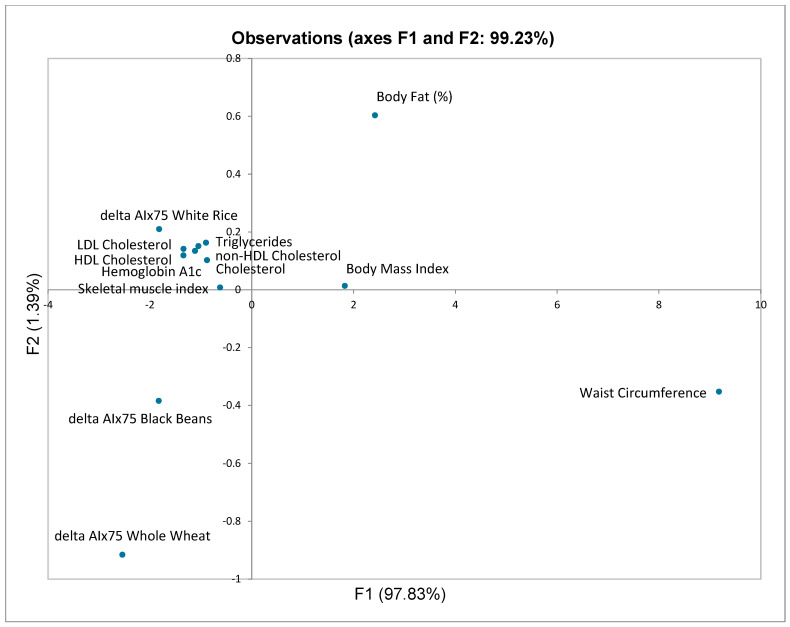
Principal Component Analysis. Abbreviations: delta AIx75, the change (T120 minus T0 min) in augmentation index normalized to a heart rate of 75 bpm.

**Table 1 nutrients-17-01159-t001:** Characteristics of the participants.

Participant Characteristics	Mean ± SD (Range)
Demographics	
Sex (male/female)	3/6
Age (years)	57 ± 6 (50–64)
Anthropometrics	
Height (cm)	170 ± 8 (158–186)
Weight (kg)	88.1 ± 13.6 (69.1–113)
Body mass index (kg/m^2^)	30.6 ± 3.7 (25.1–35.3)
Waist circumference (cm)	98.1 ± 8.2 (84–112)
Body fat (%)	36.1 ± 7.8 (26.7–47.3)
Skeletal muscle index (kg/m^2^)	7.97 ± 0.93 (6.4–9.6)
Vascular Assessments	
Brachial-ankle pulse wave velocity, left side (cm/s)	1595 ± 121 (1445–1818)
Brachial-ankle pulse wave velocity, right side (cm/s)	1564 ± 131 (1385–1766)
Systolic blood pressure (mmHg)	136 ± 10 (124–150)
Diastolic blood pressure (mmHg)	86 ± 6 (77–95)
Ankle-brachial index, left side	1.14 ± 0.07 (1.06–1.245)
Ankle-brachial index, right side	1.14 ± 0.04 (1.06–1.19)
Plasma/Serum Chemistry	
Creatinine (µmol/L)	74.3 ± 15.7 (50–93)
Aspartate aminotransferase (U/L)	18.4 ± 3.7 (14–27)
Alanine aminotransferase (U/L)	17.2 ± 7.6 (11–36)
Cholesterol (mmol/L)	5.41 ± 1.30 (3.5–7.2)
HDL Cholesterol (mmol/L)	1.37 ± 0.31 (0.9–1.8)
LDL Cholesterol (mmol/L)	3.43 ± 0.95 (1.9–4.8)
Non-HDL Cholesterol (mmol/L)	4.04 ± 1.16 (2.4–5.9)
Triglycerides (mmol/L)	1.39 ±0.62 (0.9–2.6)
Hemoglobin A1c (%)	5.59 ± 0.44 (4.8–6.4)

**Table 2 nutrients-17-01159-t002:** Peripheral hemodynamic parameters at time 0 (T0) and 120 min (T120) after food consumption.

Peripheral Hemodynamic Parameters ^1^						
**Food**	**Wheat**		**Bean**		**Rice**	
**Time**	**T0**	**T120**	**T0**	**T120**	**T0**	**T120**
Systolic pressure (mmHg)	137.6 ± 2.3	143.6 ± 4.0	140.4 ± 2.5	143.9 ± 2.2	140.8 ± 2.2	147.0 ± 1.9 *
Diastolic pressure (mmHg)	88.1 ± 2.2	87.4 ± 3.3	85.1 ± 2.0	88.0 ± 2.0 *	88.6 ± 2.3	89.4 ± 2.7
Peripheral pulse pressure (mmHg)	49.4 ± 1.9	56.1 ± 3.3 *	55.3 ± 1.8	55.9 ± 2.6	52.2 ± 2.7	57.6 ± 1.8 *
Mean arterial pressure (mmHg)	103.2 ± 2.1	102.7 ± 3.1	101.8 ± 1.9	103.8 ± 1.6	104.4 ± 2.1	106.1 ± 2.6

^1^ Values are means ± SEM. An asterisk (*) indicates that the T120 value for a food is significantly different (*p* < 0.05) from its T0 value.

**Table 3 nutrients-17-01159-t003:** Aortic hemodynamic parameters at time 0 (T0) and 120 min (T120) after food consumption.

Aortic Hemodynamic Parameters ^1^						
Food	Wheat		Bean		Rice	
Time	T0	T120	T0	T120	T0	T120
Systolic pressure (mmHg)	126.7 ± 2.2	131.4 ± 3.5	129.1 ± 2.0	133.2 ± 2.0	130.3 ± 1.7	136.0 ± 1.8 *
Diastolic pressure (mmHg)	88.7 ± 2.2	87.9 ± 3.2	86.3 ± 2.0	88.8 ± 2.0	89.0 ± 2.2	90.2 ± 2.7
Pulse pressure (mmHg)	38.0 ± 1.9	43.6 ± 2.9	42.8 ± 1.5	44.4 ± 2.7	41.3 ± 2.1	45.8 ± 1.8 *
Mean arterial pressure (mmHg)	103.2 ± 2.1	102.7 ± 3.1	101.8 ± 1.9	103.8 ± 1.6	104.4 ± 2.1	106.1 ± 2.6
Heart rate (bpm)	67.7 ± 1.9	58.2 ± 1.8 *	63.2 ± 1.5	58.0 ± 1.4 *	63.3 ± 2.1	58.0 ± 2.2 *

^1^ Values are means ± SEM. An asterisk (*) indicates that the T120 value for a food is significantly different (*p* < 0.05) from its T0 value.

**Table 4 nutrients-17-01159-t004:** Central hemodynamic parameters at time 0 (T0) and 120 min (T120) after food consumption.

Central Hemodynamic Parameters ^1^						
Food	Wheat		Bean		Rice	
Time	T0	T120	T0	T120	T0	T120
Heart rate (bpm)	67.7 ± 1.9	58.2 ± 1.8 *	63.2 ± 1.5	58.0 ± 1.4 *	63.3 ± 2.1	58.0 ± 2.2 *
Period (ms)	891.6 ± 25.1	1039.1 ± 31.7 *	953.3 ± 22.6	1038.2 ± 26.2 *	956.6 ± 31.4	1045.2 ± 38.5
Ejection duration (ms)	293.6 ± 6.1	320.4 ± 7.8 *	310.4 ± 5.2	326.1 ± 6.0 *	314.7 ± 5.8	326.1 ± 6.1
Ejection duration (%)	33.2 ± 0.8	30.9 ± 0.7	32.7 ± 0.8	31.4 ± 0.7 *	33.3 ± 0.8	31.4 ± 0.9
Aortic T2 (ms)	212.9 ± 5.0	220.7 ± 2.2	221.6 ± 4.7	227.7 ± 4.8 *	222.9 ± 3.7	229.8 ± 3.9
P1 Height (P1-DP) (mmHg)	30.3 ± 1.4	35.4 ± 2.6 *	33.1 ± 1.2	34.6 ± 1.7	31.3 ± 1.7	35.2 ± 1.2 *
Aortic augmentation pressure (mmHg)	11.6 ± 1.9	10.4 ± 1.1	13.2 ± 1.9	13.8 ± 2.0	13.1 ± 1.4	14.7 ± 1.7
Aortic Alx (AP/PP) (%)	29.4 ± 3.9	24.3 ± 2.1	30.2 ± 3.7	29.8 ± 2.8	31.8 ± 2.9	31.4 ± 2.8
Aortic Alx (P2/P1) (%)	130.7 ± 3.5	125.9 ± 2.1	130.6 ± 3.5	130.6 ± 2.9	132.2 ± 2.9	132.2 ± 2.9
Aortic Alx (AP/PP) @HR75 (%)	26.1 ± 3.6	16.2 ± 2.0 *	24.6 ± 3.4	21.6 ± 2.2	26.2 ± 3.1	23.2 ± 2.6
Buckberg SEVR (%)	170.6 ± 7.0	177.6 ± 6.0	166.6 ± 5.8	173.3 ± 5.2	166.1 ± 5.1	173.8 ± 5.6
Pressure-time index (Systole) (mmHg.s/min)	2309.0 ± 96.0	2232.6 ± 79.7	2303.9 ± 82.3	2289.0 ± 58.2	2363.3 ± 79.8	2342.2 ± 99.8
Pressure-time index (Diastole) (mmHg.s/min)	3889.3 ± 84.8	3941.2 ± 131.1	3802.6 ± 56.0	3860.2 ± 97.9	3898.3 ± 63.7	4022.9 ± 59.4 *
End systolic pressure (mmHg)	119.6 ± 2.0	120.7 ± 3.8	119.9 ± 1.7	122.6 ± 1.5	121.9 ± 1.7	125.3 ± 1.9 *
Mean arterial pressure (Systole) (mmHg)	116.0 ± 2.0	120.1 ± 3.2	117.2 ± 1.9	121.0 ± 1.8	118.8 ± 1.6	123.7 ± 1.9 *
Mean arterial pressure (Diastole) (mmHg)	96.8 ± 2.2	95.2 ± 3.3	94.3 ± 1.8	96.0 ± 1.5	97.2 ± 2.2	98.2 ± 2.5

^1^ Values are means ± SEM. An asterisk (*) indicates that the T120 value for a food is significantly different (*p* < 0.05) from its T0 value. Abbreviations: AIx, augmentation index (calculated as augmentation pressure divided by pulse pressure); AP, augmentation pressure (calculated as P2-P1); Buckberg SEVR, Buckberg subendocardial viability ratio; DP, diastolic pressure; HR75, heart rate at 75 beats per minute; P1, the incident pressure waveform during systole; P2, the reflected wave added to the incident waveform; PP, pulse pressure.

**Table 5 nutrients-17-01159-t005:** Wave reflection parameters at time 0 (T0) and 120 min (T120) after food consumption.

Wave Reflection Parameters ^1^						
Food	Wheat		Bean		Rice	
Time	T0	T120	T0	T120	T0	T120
Forward pulse height (Pf) (mmHg)	28.2 ± 1.6	28.9 ± 1.6	30.1 ± 1.4	29.0 ± 1.3	29.3 ± 1.5	31.6 ± 1.1
Reflected pulse height (Pb) (mmHg)	17.0 ± 0.9	18.1 ± 0.8	19.1 ± 0.8	19.4 ± 1.1	18.2 ± 0.9	20.2 ± 1.0
Reflection magnitude (%)	61.0 ± 2.9	63.3 ± 2.3	64.6 ± 3.7	67.3 ± 2.4	62.8 ± 2.1	64.1 ± 1.7

^1^ Values are means ± SEM. There were no significant differences between T0 and T120 values for a food.

## Data Availability

Data described in the manuscript will be made available upon request pending application and approval in accordance with ethical guidelines.

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
