# Peer review of "Distinct Effects of Wheat and Black Bean Consumption on Postprandial Vascular Responses in People with Arterial Stiffness: A Pilot Randomized Cross-Over Study"

_nutrients, 2025, doi:10.3390/nu17071159_

Round 1

Reviewer 1 Report

Comments and Suggestions for Authors

The work is well conducted, the methods used are appropriate and relatively innovative. The data analysis is thorough. The conclusions are congruent with the methods used.
The discussion appears too verbose, and needs revision and reduction.

Author Response

The work is well conducted, the methods used are appropriate and relatively innovative. The data analysis is thorough. The conclusions are congruent with the methods used.
The discussion appears too verbose, and needs revision and reduction.

We thank the reviewer for their positive comments. While we appreciate that reducing the length of the Discussion is a reasonable request, the other Reviewers have requested changes which required additional points to be added to the Discussion. For this reason it has not been possible to reduce the length of the Discussion.

Reviewer 2 Report

Comments and Suggestions for Authors

To evaluate the postprandial vascular responses of eating pulses and whole grains, this author conducted a single-blinded, controlled randomized crossover clinical trial. From this study this author found that eating whole wheat or beans acutely improved overall vascular and heart health when compared to white rice. I thought this manuscript contains newly informative knowledge. However, I also thought there are many problems that should be solved before to be published. I made comments about those problems.

Study design

  1. Eating speed might strong influence on present results.

Fast eater should have steeper and higher postprandial glucose spike than slow eater [Nutrients. 2021]. Postprandial glucose spike could influence on endothelium [Diabetes Res Clin Pract. 2008] [Int J Clin Pract. 2007]. Since eating whole wheat and beans needs might needs the number of chews much more than eating while rice, present results could be explained only by eating speed.

Kanbay M, Guler B, Ertuglu LA, Dagel T, Afsar B, Incir S, Baygul A, Covic A, Andres-Hernando A, Sánchez-Lozada LG, Lanaspa MA, Johnson RJ. The Speed of Ingestion of a Sugary Beverage Has an Effect on the Acute Metabolic Response to Fructose. Nutrients. 2021 Jun 2;13(6):1916. doi: 10.3390/nu13061916.

Glucose "peak" and glucose "spike": Impact on endothelial function and oxidative stress.

Ceriello A, Esposito K, Piconi L, Ihnat M, Thorpe J, Testa R, Bonfigli AR, Giugliano D.

Diabetes Res Clin Pract. 2008 Nov;82(2):262-7. doi: 10.1016/j.diabres.2008.07.015.

Postchallenge hyperglycaemic spike associate with arterial stiffness.

Huang CL, Chen MF, Jeng JS, Lin LY, Wang WL, Feng MH, Liau CS, Hwang BS, Lee YT, Su TC. Int J Clin Pract. 2007 Mar;61(3):397-402. doi: 10.1111/j.1742-1241.2006.01227.x.

  1. The clinical characteristic of present target population also should be taken into consideration. Difference in basal metabolic rate, difference in muscle mass, difference in capacity to produce insulin also could influence on present results.

  1. How is the influence of medication on present study? Even present study is cross-over study, the magnitude of glucose spike could influenced by some type of medication.

Abstract

  1. Many important information is lacking in present abstract.

The information about target population is necessary to be described.

   Number of target population, gender, and age is important information.

 What statistical method are used to lead present results is also should be shown.  

         The magnitude of the difference between target factors also should be shown.

Introduction

  1. Even this author described about the utility of evaluating the arterial stiffness those descriptions are chronic arterial stiffness. Since present study evaluates the postprandial vascular responses, this author should describe about the utility of evaluating postprandial arterial stiffness.

Method

  1. What methods were used to validate present study model should be clarified in method section.

  1. Eating speed should be taken into consideration in present study model.

Author Response

To evaluate the postprandial vascular responses of eating pulses and whole grains, this author conducted a single-blinded, controlled randomized crossover clinical trial. From this study this author found that eating whole wheat or beans acutely improved overall vascular and heart health when compared to white rice. I thought this manuscript contains newly informative knowledge. However, I also thought there are many problems that should be solved before to be published. I made comments about those problems.

Thank you to the Reviewer for the positive and constructive comments, and the opportunity to enhance the clarity of the manuscript.

Study design

  1. Eating speed might strong influence on present results.

Fast eater should have steeper and higher postprandial glucose spike than slow eater [Nutrients. 2021]. Postprandial glucose spike could influence on endothelium [Diabetes Res Clin Pract. 2008] [Int J Clin Pract. 2007]. Since eating whole wheat and beans needs might needs the number of chews much more than eating while rice, present results could be explained only by eating speed.

Kanbay M, Guler B, Ertuglu LA, Dagel T, Afsar B, Incir S, Baygul A, Covic A, Andres-Hernando A, Sánchez-Lozada LG, Lanaspa MA, Johnson RJ. The Speed of Ingestion of a Sugary Beverage Has an Effect on the Acute Metabolic Response to Fructose. Nutrients. 2021 Jun 2;13(6):1916. doi: 10.3390/nu13061916.

 Glucose "peak" and glucose "spike": Impact on endothelial function and oxidative stress. Ceriello A, Esposito K, Piconi L, Ihnat M, Thorpe J, Testa R, Bonfigli AR, Giugliano D.Diabetes Res Clin Pract. 2008 Nov;82(2):262-7. doi: 10.1016/j.diabres.2008.07.015.

 Postchallenge hyperglycaemic spike associate with arterial stiffness.Huang CL, Chen MF, Jeng JS, Lin LY, Wang WL, Feng MH, Liau CS, Hwang BS, Lee YT, Su TC. Int J Clin Pract. 2007 Mar;61(3):397-402. doi: 10.1111/j.1742-1241.2006.01227.x.

Response:  Thank you to the Reviewer for raising this point about the potential role of eating speed and providing the references. We recorded the time for study food consumption (line 198) and those data are now included (lines 229-230), along with the method for statistical analyses (lines 234-235). The difference in eating time between Wheat and Rice was 3.1 minutes (8.0 vs 4.9 minutes) and the difference in eating time between Bean and Rice was 2.4 minutes (7.3 vs 4.9 minutes). There was no statistical difference in the eating time between Wheat (8.0 minutes) and Bean (7.3 minutes), however, the AIx75 was significantly reduced by wheat consumption (from 26.1% to 16.2% at 0 vs 2 h, P<0.05) but not with bean consumption (26.2% to 23.2% at 0 vs 2 h, P>0.05); thus, the difference in the AIx75 response between wheat and beans is not explained by eating time.

A limitation of the present study (now included in the Discussion on lines 498-501) is that we did not measure blood glucose or other metabolic parameters during the 2 hour test. However, we did not see a difference in blood glucose concentrations between Rice and Beans at 1 h or 2 h post-consumption in our previous study with different types of beans and postprandial vascular responses of healthy individuals (Clark et al 2021 Nutr Metab CVD 31:216-226). The Reviewer raises an important concept about a spike in blood glucose affecting the endothelium and we have now addressed this in the Discussion on lines 448-473.  Additionally, it is noted that there are several differences between the present study and the aforementioned references, including the type of “food” challenge and timeframes for assessing changes in vascular responses. We investigated consumption of whole foods and vascular responses 2 hours post-challenge, whereas Kambay et al (2021) investigated blood parameters over 2 hours after consumption of apple juice (chosen for its higher fructose to glucose ratio) at difference consumption rates (500 mL in 5 minutes versus 125 mL every 15 minutes for 2 hours), Ceriello et al (2008) assessed endothelial dysfunction in 6 hour study by increasing glycemia to 10 mmol/L for 3 h and then 15 mmol/L for 3 h via an euinsulinemic hyperglycemic clamp, and Huang et al (2007) investigated relationships between a baseline assessment of cardio-ankle vascular index (CAVI) and the blood glucose response during a 75 gram oral glucose tolerance test. Relevant information related to these articles has been incorporated into the Discussion in the aforementioned section. 

  1. The clinical characteristic of present target population also should be taken into consideration. Difference in basal metabolic rate, difference in muscle mass, difference in capacity to produce insulin also could influence on present results.

Response:  We agree that other clinical characteristics of the participants should be taken into consideration and have acknowledged this in the Discussion (lines 473-478). We did not obtain data on basal metabolic rate or insulin sensitivity and thus cannot discuss those parameters. The principal component analysis revealed that waist circumference, an indicator of visceral fat deposition, was a major factor in the post-prandial AIx75 response (Discussion, lines 374-376). The serum metabolic factors related to cholesterol, triglycerides, and glycemic control, and skeletal muscle index, were clustered with the delta AIx75 for white rice but their location near the origin of the plot suggests these metabolic factors had a minor effect on the postprandial response, and that they were not impacting the postprandial response to beans and whole wheat (Discussion, lines 376-380).

  1. How is the influence of medication on present study? Even present study is cross-over study, the magnitude of glucose spike could influenced by some type of medication.

Response:  We have specifically excluded persons taking medication that affects blood pressure or has a vasoactive effect. Both are important since they can affect vascular parameters associated with arterial stiffness. We also excluded persons with diabetes (HbA1c was ≤6.5%, Table 1), and none of the participants were taking glucose-lowering or lipid-lowering medications (Discussion, lines 475-476) which might influence the blood glucose response. 

Abstract

  1. Many important information is lacking in present abstract.

The information about target population is necessary to be described.

   Number of target population, gender, and age is important information.

 What statistical method are used to lead present results is also should be shown.  

         The magnitude of the difference between target factors also should be shown.

Response:  Additional details have been added to the Abstract (line 19) to indicate n=9 for the target population, 3 males and 6 females, with an age range or 50-64 years old. The statistical test for the baseline versus 2 h comparison is indicated on line 21. The magnitude of the changes are now indicated for blood pressure (line 22) and augmentation index at 75 bpm (line 24).

Introduction

  1. Even this author described about the utility of evaluating the arterial stiffness those descriptions are chronic arterial stiffness. Since present study evaluates the postprandial vascular responses, this author should describe about the utility of evaluating postprandial arterial stiffness.

Response:  The Introduction contains a number of references that relate to postprandial arterial stiffness as opposed to chronic arterial stiffness. Please see references 15-21 (lines 64-71). As requested by the reviewer, reference 18 (Staiculescu et al 2013) provides the mechanistic basis for examining postprandial effects on arterial stiffness, namely that regular vasoconstriction in response to meals can cause arterial remodeling, which ultimately increases stiffness. However, to clarify these points, we have expanded on this topic and added reference 17 (Fewkes et al 2022) that describes the relationship between post-meal impairment of endothelial dysfunction with subsequent arterial remodeling, particularly in relation to dietary fat consumption (lines 64-66).

Method

  1. What methods were used to validate present study model should be clarified in method section.

Response:  The main requirement for the study population (model) was the presence of arterial stiffness, and this was determined using brachial-ankle pulse wave velocity (baPWV). As described in the Methods (lines 101-102), a baPWV greater than 1400 m/s is indicative of arterial stiffness (validation described in reference 23). This point has also been added to section 2.2 of the Methods (line 120) and is present in the Discussion (line 480).

  1. Eating speed should be taken into consideration in present study model.

Response:  As requested, eating speed has been quantified and added to the Results, as well as the Discussion. This point is also addressed in the response to Item 1 above (Study Design).

Reviewer 3 Report

Comments and Suggestions for Authors

This manuscript revealed the effectiveness of food for cardiac and vascular function. I think that this manuscript will provide readers with clinically beneficial results. I have some questions. Please make clear.

1. Because PWV is the evaluation factor, is it necessary to exclude vascular disease (including PAD)? It may be a good idea to provide details about cardiovascular disease. In addition, why are patients with high blood pressure excluded?

2. How would you measure and evaluate blood pressure?

3. I think it is a little too early for effects on vascular function to appear two hours after ingestion of a food. In the method, it would be a good idea to cite previous papers and explain the research methods in detail. I also think it would be good to discuss in the discussion the mechanism by which a single administration exerts its effects in a short period of time.

Author Response

This manuscript revealed the effectiveness of food for cardiac and vascular function. I think that this manuscript will provide readers with clinically beneficial results. I have some questions. Please make clear.

Thank you to the Reviewer for the positive and constructive comments, and the opportunity to enhance the clarity of the manuscript.

  1. Because PWV is the evaluation factor, is it necessary to exclude vascular disease (including PAD)? It may be a good idea to provide details about cardiovascular disease. In addition, why are patients with high blood pressure excluded?

Response:  Yes, the presence of vascular disease was an exclusion criterion. This included PAD, and anyone with an ankle-brachial index (ABI) <0.9 (clinical definition of PAD) was excluded. We failed to mention PAD in the manuscript, and it is now included in the exclusion criteria on line 131-132, and how it is measured in given on line 179-180. Likewise, we added ABI into Table 1 (Characteristics of the participants), and the data show that ABI was normal (>0.9) for all participants. Additionally, anyone taking medication that had a vasoactive effect was excluded, and this is mentioned in the manuscript on lines 135-137. Persons with high blood pressure were excluded because they should be on medication and anyone who had high blood pressure at the screening visit would be referred to a physician for treatment.

  1. How would you measure and evaluate blood pressure?

Response:  Blood pressure was measured with an automated BP monitor. The details have been added to the Methods (line 180-182).

  1. I think it is a little too early for effects on vascular function to appear two hours after ingestion of a food. In the method, it would be a good idea to cite previous papers and explain the research methods in detail. I also think it would be good to discuss in the discussion the mechanism by which a single administration exerts its effects in a short period of time.

Response:  Vascular function changes within 2 hours have been reported in the literature. An example has been added to the Introduction [reference 19, Kume et al 2025] to illustrate this point (lines 67-69). With respect to a possible mechanism, this is presented on lines 65-66 [reference 17, Fewkes et al 2022].

Round 2

Reviewer 3 Report

Comments and Suggestions for Authors

The revised manuscript is much improved. I pay my respects for the authors have made a sincere 
effort to address concerns. I have no more question.  
